# Does Influenza Vaccination Reduce the Risk of Contracting COVID-19?

**DOI:** 10.3390/jcm11185297

**Published:** 2022-09-08

**Authors:** Francesc Alòs, Yoseba Cánovas Zaldúa, María Victoria Feijóo Rodríguez, Jose Luis Del Val Garcia, Andrea Sánchez-Callejas, Mª Àngels Colomer

**Affiliations:** 1Primary Healthcare Center, CAP Passeig de Sant Joan, Gerència Territorial de Barcelona, Institut Català de la Salut, 08010 Barcelona, Spain; 2Unitat de Suport a la Recerca Barcelona, Foundation Primary Care Research Institute Jordi Gol i Gurina (IDIAPJGol), 08025 Barcelona, Spain; 3BASIQ Unitat d’Avaluació, Sistemes d’Informació i Qualitat, Àmbit Barcelona Ciutat, Institut Català de la Salut, 08029 Barcelona, Spain; 4Department of Mathematics ETSEA, University of Lleida, 25198 Lleida, Spain

**Keywords:** COVID-19, influenza vaccines, comorbidity, health status disparities, social determinants of health, public health

## Abstract

The concurrent timing of the COVID-19 pandemic and the seasonal occurrence of influenza, makes it especially important to analyze the possible effect of the influenza vaccine on the risk of contracting COVID-19, or in reducing the complications caused by both diseases, especially in vulnerable populations. There is very little scientific information on the possible protective role of the influenza vaccine against the risk of contracting COVID-19, particularly in groups at high-risk of influenza complications. Reducing the risk of contracting COVID-19 in high-risk patients (those with a higher risk of infection, complications, and death) is essential to improve public well-being and to reduce hospital pressure and the collapse of primary health centers. Apart from overlapping in time, COVID-19 and flu share common aspects of transmission, so that measures to protect against flu might be effective in reducing the risk of contracting COVID-19. In this study, we conclude that the risk of contracting COVID-19 is reduced if patients are vaccinated against flu, but the reduction is small (0.22%) and therefore not clinically important. When this reduction is analysed based on the risk factor suffered by the patient, statistically significant differences have been obtained for patients with cardiovascular problems, diabetics, chronic lung and chronic kidney disease; in all four cases the reduction in the risk of contagion does not reach 1%. It is worth highlighting the behaviour that is completely different from the rest of the data for institutionalized patients. The data for these patients does not suggest a reduction in the risk of contagion for patients vaccinated against the flu, but rather the opposite, a significant increase of 6%. Socioeconomic conditions, as measured by the MEDEA deprivation index, explain increases in the risk of contracting COVID-19, and awareness campaigns should be increased to boost vaccination programs.

## 1. Introduction

The COVID-19 pandemic of 2019 originated in the city of Wuhan, China, at the end of December 2019. COVID-19 is an acute respiratory disease caused by the severe acute respiratory syndrome coronavirus 2 (SARS-CoV-2). It spread rapidly worldwide, posing a threat due to its virulence and the high rate of morbidity and mortality it causes [1].

Many resources are being invested in measures to alleviate, or try to stop, the spread of COVID-19. At first, it was thought that the transmission of the virus could be reduced by widespread vaccination and that herd immunity would be achieved if the vaccinated proportion of a population reached 70%. The current reality is that the rate of vaccination varies significantly depending on the economic situation of a region, against a backdrop of increasing global inequality. The percentage of people vaccinated against COVID-19 in developed countries is much higher than in underdeveloped ones, running the risk of prolonging the pandemic as new mutations appear in the developing world [2,3].

Various studies have shown that the risks of infection and transmission of the disease do not depend on whether the complete schedule of vaccines against COVID-19 has been received [4,5]. Vaccinated people have a lower risk of serious illness but are still an important part of the pandemic. The ease with which this disease is transmitted between vaccinated and unvaccinated people allows new variants to emerge, with potentially lower percentages of protection with current vaccines. There are currently 10 variants being monitored, of which two are ‘variants of concern’ [6]. The high rate of appearance of new variants generates a level of confusion at the health, social and economic levels, the repercussions of which have prompted increased scientific research into how the virus works and preventative measures and treatments that could mitigate its effects on health and the spread of the pandemic. Prestigious journals have published special issues dedicated exclusively to advances in research on SARS-CoV-2 (ASM Journals; Nature, STOTEN, JMIR, JSSE, SAGE Journals, Arab Journal of Basic and Applied Sciences, among others). Some of the published studies have looked at whether effective measures for other types of viruses can reduce the effects of SARS-CoV-2.

Influenza and COVID-19 are two viral diseases with similar modes of transmission, mainly through respiratory droplets emitted by both symptomatic and asymptomatic sufferers [7,8]. They produce viral respiratory diseases that can be clinically indistinguishable and cause serious complications, hospitalization, and death, mainly in at-risk subgroups of the population, such as the elderly and those with chronic disease [9,10]. Moreover, because they are viruses that mainly cause respiratory problems, their activity peak can occur during the same time of year, namely the winter months in temperate countries.

Annual influenza vaccination is one of the most efficient and cost-effective strategies to prevent and control influenza epidemics [11]. Most developed countries have annual seasonal influenza vaccine programs and similar means of tackling the virus [6,12,13].

Over time, the recommended age for influenza vaccination has gradually been lowered to approximately 50 years of age as a more conservative strategy to address more of the underlying chronic disease, unknown and unsuspected, above that age [14]. The World Health Organization (WHO) has identified children from six months to five years of age, pregnant women, those with chronic conditions and the elderly as high-risk groups for flu and as candidate groups for the flu vaccine to reduce infection, serious complications, and premature death [11]. In Europe, annual recommendations regarding the use of seasonal influenza vaccines vary between Member States, but there is consensus that individuals with specific underlying conditions and the elderly be vaccinated. In the USA, annual flu vaccination is recommended for all persons aged six months or over who do not have a contraindication [15].

Flu and COVID-19 affect populations unequally around the world, more severely impacting groups with lower socioeconomic status, resulting in geographic inequalities [16,17]. Deprivation indices allow the lack of economic and social resources to be quantified, making it possible to rank countries from the least to the most deprived. They are constructed by combining various, weighted, socioeconomic indicators, using a multivariate statistical model. They are widely used tools in the analysis of health inequalities. Deprivation indices assess the difficulty of access to material and social resources (education, work, housing, food, culture, and social development) at levels that are considered acceptable in a society. This expanded perspective goes beyond purely monetary aspects and is in line with holistic models of health.

Vaccination campaigns have been important in the COVID-19 pandemic, both to protect healthcare systems and the health of the population [18,19]. International health organizations have been forceful in maintaining established immunization programs [20,21]. Most health professionals have spoken out in favor of strengthening flu vaccination programs, arguing that increased flu vaccine coverage could help improve the treatment of COVID-19 patients, facilitate diagnosis and reduce pressure on healthcare systems, both in primary care centers and at the hospital level, especially in intensive care units (ICUs) [19]. The reality of the 2020–2021 campaign has been to effectively ignore flu, although in general, the population measures taken by governments and health services to protect against SARS-CoV-2 have also protected against flu. However, given the apparent difficulty of eradicating SARS-CoV-2, it is expected that coming campaigns will have to address the coexistence of both viruses.

The first Omicron variant from South Africa was notified to the WHO on 24 November 2021. According to some researchers [22], the Omicron variant of the virus causing COVID-19 probably acquired at least one of its mutations by picking up a fragment of genetic material from another virus, possibly one causing the common cold, which was present in the same infected cells. This variant presents symptoms very similar to that of common flu and has a large number of mutations, some of which are worrying, and faster rates of spread than previous variants that have been detected. If the research finally confirms the relationship between Omicron and the flu virus, it becomes especially important to protect the population from the flu virus as well.

Some studies have suggested that being vaccinated against flu could confer some protection against SARS-CoV-2 infection, leading to reductions in hospitalizations, ICU admissions and mortality [23,24]. Most studies have been of the general population and based on small samples [25]. Not all studies have found a significant association between the flu vaccine and the risk of infection from SARS-CoV-2. Some authors [26] found a partial relationship and/or a reduction in the risk of infection for people under 65 years of age, while others [27] have shown a stronger association in the subgroup of people older 64 years old. Findings related to the possible relationship between the flu vaccine and SARS-CoV-2 infection are insufficient and contradictory. Additional studies to better quantify the relationship are needed to improve understanding of the possible protective role of the flu vaccine on the spread of COVID-19.

The main objective of this study was to determine the association between the seasonal flu vaccine and the risk of contracting COVID-19 in those patients for whom the seasonal flu vaccine is recommended, considering their risk level and the probability of serious complications. This relationship should be observable in the different groups at high risk of flu complications, including the subgroup of people over 60 years of age who do not present any other risk factor for flu complications. A further objective was to assess the probability of contracting COVID-19 in people vaccinated against flu in relation to different levels of deprivation.

## 2. Materials and Methods

### 2.1. Data

Our database comprised all patient records in the high-risk groups for flu complications who were recommended flu vaccination in the 2020–2021 vaccination campaign in the primary care units of the Gerencia Territorial de Barcelona (GTB), Institut Català de la Salut (ICS), between 15 October 2020 and 15 March 2021. Patients with a history of contracting COVID-19 before 15 October 2021 were excluded from the database, as well as those with a registration of COVID-19 before the date of their flu vaccination.

Institutionalized patients (5191 people living in nursing homes) were studied separately, because their conditions, environment and habits were different from the rest of the 429,537 patients (193,072 patients and 236,465 patients, vaccinated and not vaccinated against flu, respectively). The flu vaccines administered were the trivalent vaccine that included the B virus represented B/Colorado/06/2017 (lineage B/Victoria/2/87), A/Brisbane/02/2018 (H1N1) pdm09 and A/Kansas/14/2017 (H3N2)—and the quadrivalent vaccines which included the above together with the B/Phuket/3073/2013 strain (lineage B/Yamagata/16/88).

The information related to flu vaccination and diagnosis of COVID-19 came from the computerized clinical patient history records. “COVID-19 infection was diagnosed mostly by real-time PCR (RT-PCR). In a few cases we cannot discard that it was diagnosed by rapid antigen test”. The data were stored in a pseudonymized database containing age, gender, risk factor for flu vaccination, flu vaccine receipt, incidence of COVID-19 and the MEDEA deprivation index. The MEDEA deprivation index (Mortality in small Spanish areas and Socioeconomic and Environmental Inequalities) has been used in Spain since 2007 to analyze health inequalities in small parts of urban areas, and for the identification and prioritization of territories of special socioeconomic vulnerability [28].

The sample size used was larger than strictly necessary to detect clinically significant differences. In many cases, the inference tests that were used can give statistically significant differences that are not clinically important.

### 2.2. Statistical Analysis

The data analysis was divided into three phases. In the first phase, the global level was analysed to detect any significant statistical differences between the risk of SARS-CoV-2 infection between people vaccinated against flu and those not vaccinated, according to age. Next, we looked to see whether the risk factor for flu complications determined the effect of the flu vaccine on the risk of contracting COVID-19. Finally, we examined whether a patient’s level of deprivation affected either the probability of being vaccinated against flu and/or the risk of contracting COVID-19.

A descriptive study was carried out to estimate the proportion of people infected with SARS-CoV-2 who were vaccinated against flu. To study whether these differences were statistically significant, a unilateral proportions test was performed to analyze whether the flu vaccine could reduce the risk of contracting COVID-19.

All statistical analyses were performed using the R Core package [29].

## 3. Ethical Considerations

This study followed the standards of Good Clinical Practice and the principles of the Declaration of Helsinki. The project had the approval of the Clinical Research Ethics Committee of the Institut d’Investigació en Atenció Primària (IDIAP) Jordi Gol (reference 20/242-PCV). The confidentiality and anonymity of the patient data included in the study conformed with Regulation 2016/678 of the European Parliament and of the Council of 27/2016 on Data Protection and Organic Law 3/2018.

## 4. Results

### 4.1. Relationship between Vaccination against Flu and SARS-CoV-2 Infection

Of the 429,537 people in the database, 44.95% were vaccinated against flu and 56.80% were women (Table 1). The percentage of people vaccinated against flu varied with age, with people over 80 years of age being most likely to be vaccinated (61.92%). The percentage of women vaccinated against flu was 56.80%; on average 12 percentage points higher than men, tin all age groups there are more women vaccinated than men, except for under 15 and above 80 years. 

On average, people contracting COVID-19 represented 4.23% of those vaccinated against flu and 4.45% of those not vaccinated (Figure 1). Globally, the percentage of people contracting COVID-19 was higher in people not vaccinated against flu, this difference being statistically significant (*p* < 0.001) but not clinically important (odds ratio 0.91, 95% CI 0.89–0.94; *p* < 0.001). When this percentage was calculated by age, lower values of COVID-19 incidence were not observed in all cases for those vaccinated against flu (Figure 1). For those under 60 years of age the difference was not statistically significant. However, for those over 60, the percentage of those vaccinated against flu who contracted COVID-19 was higher than in the unvaccinated group, the difference being statistically significant (Table 2), although these statistically significant differences were not clinically important.

### 4.2. SARS-CoV-2 Infection Levels According to Flu Risk Group and Vaccination against Flu

In most age groups, the higher the number of different risk factors for flu complications that a person presents, the higher the probability that they will have been vaccinated against flu (Figure 2). This can mask the effect of the flu vaccine on the risk of contracting COVID-19, since these groups are those most susceptible to SARS-CoV-2 infection.

When the relative risk of contracting COVID-19 was examined in relation to the various risk factors for flu complications (Table 3), the differences observed were not statistically significant for most factors. For those factors that did show significant effects, the differences were small and not clinically significant. Statistically significant differences were observed for patients with chronic cardiovascular diseases, diabetes mellitus and lung and kidney problems; these patients account for 80% of the sample. When stratifying by age, in each one of these groups no differences are observed between those vaccinated and those not vaccinated against flu

We also examined whether there were significant differences in the risk of contracting COVID-19 according to the number of risk factors for flu complications presented by each patient (Table 4). The differences observed were not statistically significant, except for patients who only presented with a single risk factor. At the mean level of risk factors, 0.7% more unvaccinated people contracted flu, and this value was clinically significant.

### 4.3. Relationships between the MEDEA Deprivation Index, Flu Vaccination and SARS-CoV-2 Infection

The MEDEA index values were grouped into percentiles from 10 to 100. First, we examined whether the proportion of people vaccinated against flu and contracting COVID-19 varied depending on the level of deprivation as measured by the MEDEA index. The percentages of people with high risk factor scores for flu complications vaccinated against flu and contracting COVID-19 were similar for the different MEDEA index values, except in the case of high MEDEA values, where the probability of being vaccinated was lower and the risk of presenting with COVID-19 was higher (Figure 3).

The proportion of people contracting COVID-19 at different MEDEA index values, according to whether they were vaccinated against flu (Figure 4), was higher when they were not vaccinated against flu and at higher MEDEA index values. The differences were statistically significant (*p* < 0.001) and clinically important.

The probability of presenting with COVID-19 increases with increasing MEDEA index value (Figure 5). People under 60 years of age showed higher COVID-19 incidence rates than people 60 years of age or older, at high MEDEA index values.

## 5. Discussion

In general, it was observed that people vaccinated against flu had a lower risk of presenting with COVID-19 than those who had not been vaccinated (4.23% vs. 4.45%, respectively), but the difference was not clinically significant. Analyzing the data by age, there was no consistent reduction in the probability of presenting with COVID-19 in those vaccinated against flu: in two of the age ranges studied there was a reduced risk of contracting COVID-19: in four age ranges there was an increase; and in three age ranges the differences were practically nil.

The odds ratio of contracting COVID-19 in patients who have been vaccinated against flu was 0.91, far from the value of 0.76 obtained by Conlon et al. (2021) [30] in a group of 4.5 million patients from Michigan. In our study, the sample consisted of patients with a high-risk of flu complications, who were recommended to take vaccination against flu, while in the sample examined by Conlon et al. the sample group was not restricted to any condition or risk group for flu. In most cases, people who are at higher risk of complications from flu are also at higher risk of contracting COVID-19, and could take stronger measures to protect themselves from COVID-19, unlike the population without risk factors.

In a sample of 6856 employees of a Dutch hospital, Debisarun et al. (2021) [31] observed relative reductions in the risk of contracting COVID-19 of 37% and 49% after flu vaccination during the first and second waves of the COVID-19 pandemic, respectively, with values like the 38% found by Vila Córcoles et al. (2020) [23]. The reduction in the global risk of contracting COVID-19 after flu vaccination in our study was only 9%.

The influenza vaccines included in this study were mainly trivalent and quadrivalent inactivated influenza vaccinations following the recommended composition of influenza virus vaccines for use in the 2020–2021 northern hemisphere influenza season [32] as in other studies [31,33]. The differences between the other studies that point to a protective effect could be due to the representation of our sample of patients. The sample is more homogeneous than other studies, and all the patients have some risk factor for flu complications. Most of the studies were not restricted to any condition or risk group for flu or age and could overestimate the effect because people who were vaccinated may be healthier in general than those who were not vaccinated [30,31,34].

Patients with comorbidities such as high blood pressure, diabetes mellitus, chronic obstructive pulmonary disease, heart disease, neoplasms and HIV have a higher risk of contracting COVID-19 [35]. Patients with chronic obstructive pulmonary disease develop substantial and severe symptoms and suffer comparatively higher mortality rates. Analyzing the effect of the flu vaccine based on the risk factors for flu complications, we observed reductions in the risk of contracting COVID-19 for patients with cardiovascular diseases (0.6%), pulmonary diseases (0.6%), diabetes (0.5%) and chronic kidney disease (0.4%), results which corroborate those of Vila Córcoles et al. (2020) [23]. These four risk factors for flu complications account for more than 80% of the total population at risk. Reducing the risk of contracting COVID-19 in these patient groups is important because they have a higher risk of mortality from COVID-19 [36]. No statistically significant differences were observed between the rest of the risk factors for flu complications. The most prevalent risk factors for flu complications have strict control and monitoring programs that are likely to reduce the positive effect of the flu vaccine.

When only one risk factor is present, if the patient was vaccinated for flu the probability of contracting COVID-19 decreased by 0.7%, while no differences were observed in cases presenting more than one risk factor. The number of risk factors increased with age, and as the risk factors increased the probability of presenting with COVID-19 increased, with no statistically significant differences being observed between those vaccinated against flu and those not vaccinated. This result could be explained by the fact that an increase in risk factors increases a patient’s susceptibility to contracting or becoming infected with other pathologies.

Nursing homes for the elderly are spaces that facilitate the spread of contagious diseases [37,38]. The proportion of institutionalized patients contracting COVID-19 increased significantly in the case of those vaccinated for flu (12.6%) compared to those not vaccinated (6.6%). This could be explained by the fact that institutionalized people are generally those at high-risk and have multiple risk factors for flu complications, as well as for the SARS-CoV-2 infection itself and live in an environment which favors the spread of respiratory pathogens [39]. 

Various studies have evaluated the effectiveness of the flu vaccine in people over 65 years of age and concluded that the effect may be mild or moderate in preventing flu-like illnesses [38,40]. Flu outbreaks occur even in nursing homes with high vaccination rates in years when there is little difference between the circulating strain and the vaccine, raising questions about the efficacy of the flu vaccine, especially in frail elderly people who may not be able to acquire an adequate immune response, which might explain the results obtained in our study.

Conditions in nursing homes for the elderly increase contact and interaction between residents, increasing the speed of spread of SARS-CoV-2. The age of institutionalized patients, together with the day-to-day routines in nursing homes, means that this group of patients behave and respond differently from others when protective measures are applied [37]. Likewise, patients who live in nursing homes have vulnerability characteristics that provide a favorable environment for the rapid spread of the flu virus and other respiratory pathogens [38]. Infections can be introduced by new or transferred staff, visitors or residents, and outbreaks of flu in such settings can have devastating consequences for residents, as well as place additional pressures on health services. In the coming decades, the population trend is for the number of elderly people to increase, so it is likely that the number of residences for the elderly will also increase. The need to improve management, prevention and control of infectious diseases will therefore become especially important if the impact of new and existing contagious diseases is to be minimized. This study reinforces the importance of promoting changes in nursing homes and in reinforcing general measures, including vaccination against both flu and COVID-19, not only in patients institutionalized in nursing homes, but also in staff and visitors.

The COVID-19 pandemic has had a range of impacts, and populations with the poorest socioeconomic status have been the most affected. Populations with the greatest deprivation (measured according to the MEDEA index) have a lower percentage of flu vaccination and a greater probability of presenting with COVID-19, apparently consistent with results published by some authors [41,42,43,44,45]. This relationship may be since the highest levels of deprivation include the worst socioeconomic conditions, housing characteristics, types of employment and access to education and culture. In addition, deprived populations tend to neglect population vaccination programs, thus increasing their probability of contracting COVID-19. Deprivation is clearly a very important factor in explaining the unequal percentages of SARS-CoV-2 infection and the implementation of flu vaccination programs in certain urban areas or geographic regions.

The COVID-19 pandemic has caused both hospital and primary care medical services to become overburdened all over the world. The continuation of effective immunization programs is essential to protect people against diseases and outbreaks that can be prevented using vaccines, and so reduce the burden of respiratory diseases during the period in which both COVID-19 and flu may overlap. This study also reinforces the importance of preventive measures against both diseases at the same time. The results are especially important in the light of the probability that we will have to live with both viruses for a long time to come.

Future research needs to study the potential impact of nasally administered influenza vaccine to sufficiently stimulate innate local immunity in the respiratory tract against other respiratory viruses, including SARS-CoV-2, which is the primary infection site for SARS-CoV-2. In this way, the local respiratory immune system would provoke an intense and rapid response that would make other respiratory viral infections difficult [46].

## 6. Conclusions

Patients in high-risk groups for flu complications vaccinated against flu have a lower risk of contracting COVID-19. The differences are statistically significant, but not very relevant from the clinical point of view. A reduced risk of contracting COVID-19 has been observed in patients with cardiovascular disease, lung disease, chronic kidney disease and diabetes. These four risk factors for flu complications account for more than 80% of the total population at risk. Reducing the risk of contracting COVID-19 in these groups of patients is important because they are patients with a higher risk of mortality from COVID-19. Finally, we observed that increasing levels of socioeconomic deprivation, as measured by the MEDEA index, increase the risk of contracting COVID-19 in populations not vaccinated against flu and in patients with a higher risk of contracting COVID-19.

## Figures and Tables

**Figure 1 jcm-11-05297-f001:**
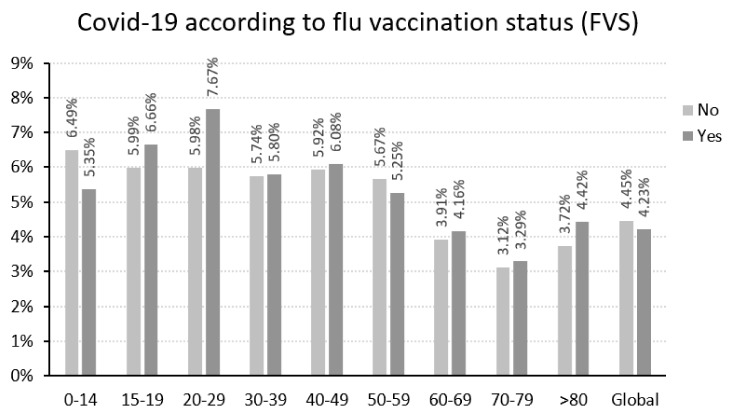
Probability of contracting COVID –19 depending on whether the patient had received the flu vaccine.

**Figure 2 jcm-11-05297-f002:**
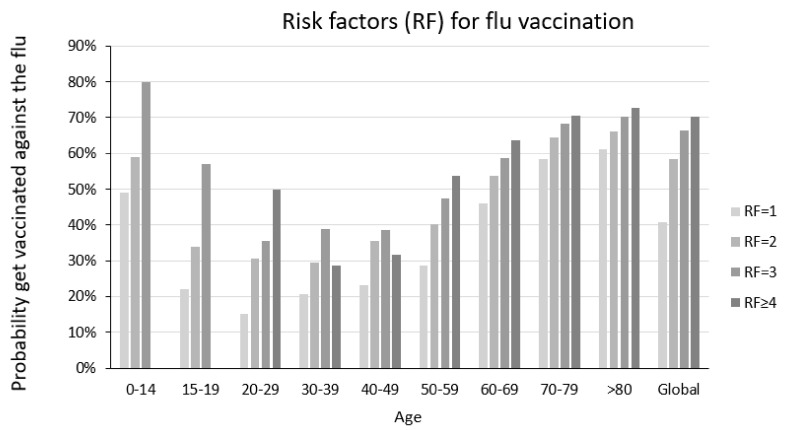
Proportion of people vaccinated against flu according to age and the number of risk factors for flu complications.

**Figure 3 jcm-11-05297-f003:**
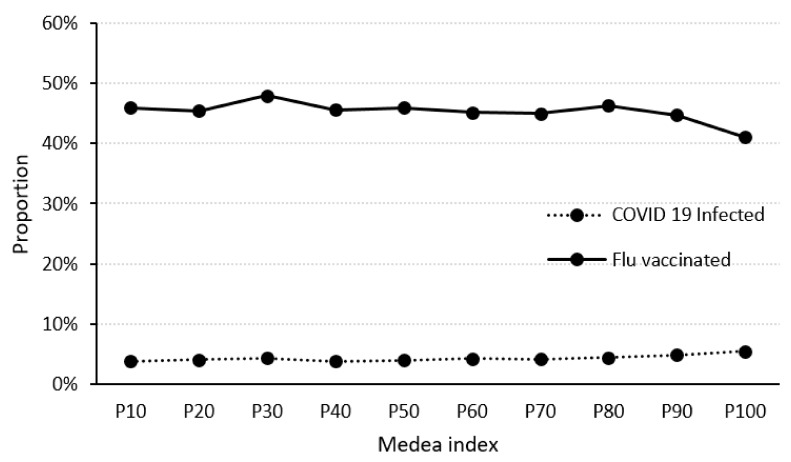
Relationship between the *p* of people vaccinated against flu and presenting with COVID-19, and the MEDEA index.

**Figure 4 jcm-11-05297-f004:**
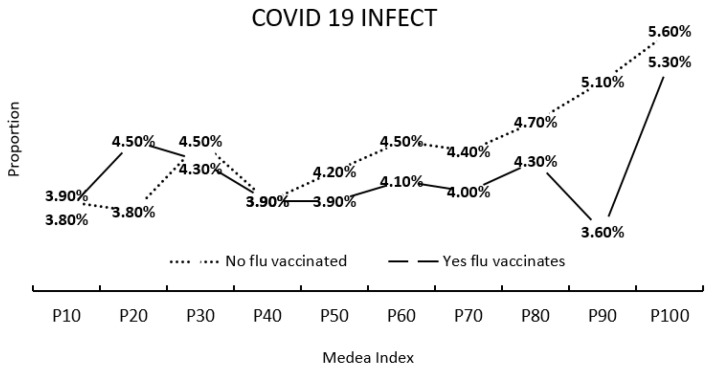
Proportion of people contracting COVID-19 according to whether or not they had been vaccinated against flu, and the MEDEA index.

**Figure 5 jcm-11-05297-f005:**
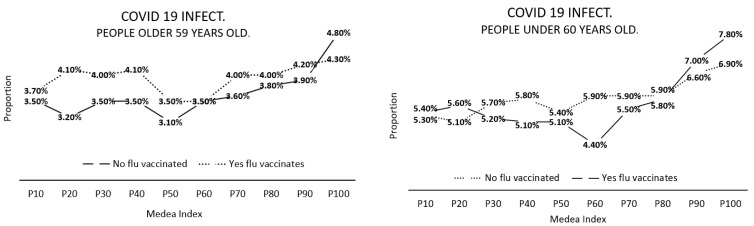
Probability of contracting COVID-19 depending on whether or not a person has been vaccinated against flu, and the size of the MEDEA index. For people above and below 60 years of age.

**Table 1 jcm-11-05297-t001:** Percentage of people contracting COVID –19 who were vaccinated against flu, according to age and sex.

Age	Sample Size	Vaccinated against Flu (%)	Vaccinated Women (%)
0–14	8348	49.4	48
15–20	5578	21.3	23.2
21–30	12,540	16.1	21
31–40	18,991	21.9	27.2
41–50	31,466	25	28.1
51–60	58,278	31.5	33.9
61–70	122,390	43.1	43.6
71–80	100,797	58	56.8
>80	71,149	61.9	59.8
Total	429,537	44.95	56.8

**Table 2 jcm-11-05297-t002:** Probability of SARSCoV-2 infection, based on flu vaccination status (FVS), stratified by age (over or under 60 years of age).

	Probability of SARS-CoV-2 Infection		*p*-Value
	No FVS	Yes FVS	
Total population	4.45%	4.23%	<0.001
<60 years old	5.83%	5.70%	0.3585
≥60 years old	3.62%	3.72%	<0.001

**Table 3 jcm-11-05297-t003:** Percentage of people contracting COVID –19 according to whether they had been vaccinated for flu (FVS). *p*-value of the proportion comparison test.

Risk Factor	Sample Size	Proportion of People Contracting COVID-19 Not FVS/FVS (*p*-Value)		Relative Risk	
		All	>59 Years Old	All	>59 Years Old
Chronic cardiovascular diseases	231,853	4.6/4 (<0.001)	3.9/3.8 (0.248)	0.87	0.97
Digestive problems	8699	5.1/4.6 (0.2699)	4.2/4.3 (0.8851)	0.90	1.02
Mellitus diabetes	75,986	5/4.5 (<0.001)	4.5/4.2 (0.0615)	0.90	0.93
Pregnant women	3688	5.8/6.5 (0.3799)		1.12	
Immunodeficiencies	6882	5.2/4.5 (0.2094)	3.2/4 (0.2416)	0.87	1.25
Malformations	1135	3.5/4.8 (0.2971)	4.2/2.3 (0.4831)	1.37	0.55
Chronic kidney disease	34,597	4.7/4.3 (0.0457)	4.6/4.2 (0.0967)	0.91	0.91
Neoplasms	4926	4.9/4.1 (0.1535)	4/4 (0.9901)	0.84	1.00
Neurological diseases	6525	4.5/4.7 (0.8261)	4.8/4.7 (0.8706)	1.04	0.98
Chronic lung diseases	92,187	5.4/4.8 (<0.001)	4.2/4.4 (0.4177)	0.89	1.05
Hematologic disorders	6477	5.2/4.3 (0.0826)	4.2/3.8 (0.5704)	0.83	0.90
Organ transplant	1668	6.1/5.4 (0.5284)	5.1/4.7 (0.7882)	0.89	0.92
HIV infection	2245	2.8/3.2 (0.6069)	2.3/2.8 (0.7066)	1.14	
Linving in a nursing home	5191	------	6.6/12.6 (<0.001)		1.91
Older than 60 years without other RF.	98,570	3.3/3.4 (0.3701)		1.03	
Any age with RF different from the average for their age	330,967	4.9/4.2 (<0.001)	3.9/3.8 (0.1139)	0.86	0.97

**Table 4 jcm-11-05297-t004:** Percentage of people contracting COVID-19, depending on the number of risk factors for flu complications, according to whether they had been vaccinated against flu (FVS).

Number of Risk Factors for Flu Complications	Probability of SARS-CoV-2 Infection		*p*-Value
	No FVS	Yes FVS	
0	3.3	3.4	0.3701
1	4.9	4.2	<0.001
2	4.5	4.4	0.6594
3	5.0	5.2	0.5417
≥4	7.2	6.7	0.4702

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
