# Peer review of "Does Influenza Vaccination Reduce the Risk of Contracting COVID-19?"

_jcm, 2022, doi:10.3390/jcm11185297_

Round 1

Reviewer 1 Report

Congratulations to the authors, the work is well structured, scientifically valid and interesting. I also think that the presentation of the results, their description and the final discussion are done in an excellent way.

I think the study is interesting since it shows how in subjects vaccinated for influenza the vaccine does not induce protection against Covid-19, although this may seem obvious in many respects it can nevertheless serve to underline the importance of specific vaccination for SARS. -CoV-2. Consequently, I have no big requests but only minor comments and suggestions.

To make the work more complete and attractive the following information could be added.

1)      Specify with which type of test the positivity / negativity to Covid-19 was verified. For example Molecular swab or rapid antigen test and / or research of anti-spike or anti-Nucleocapsid antibodies, in the subjects studied. In particular, the search for antibodies to SARS-CoV-2 should be indicated. A few lines in the materials and methods section are enough.

2)      Also in the Methods section it should be indicated in my opinion if, for the subjects included in the study, they were subjected to other vaccinations in the proximity of the flu vaccination (for example because of pediatric subjects or for prophylactic purposes in relation to any trips abroad). I would say to consider a period of three months before and three months after the flu shot. Most likely the percentage will be very low, but for the sake of completeness you could indicate if the subjects have undergone other vaccinations (yes or no), if yes how many (ie in what percentage) for each age group considered.

3)      In table 1 the authors show the percentage of subjects who have had Covid-19 and vaccinated for the flu. While in table 3 the authors show the frequency of subjects who have developed Covid-19, depending on the presence of comorbidities and other pathologies, in relation to the anti-flu vaccination. The authors could add an analysis that summarizes and extends the data reported in table 1 and table 3. The authors could analyze the percentage of subjects who developed Covid-19, despite the anti-flu vaccination, in relation to the presence of other pathologies (diabetes, cardiovascular diseases, immunodeficiencies, etc.) showing the data divided by age groups. This for highlight if there is a difference (between vaccinated and unvaccinated subjects) in the development of Covid-19 in relation to chronic / primary diseases and to the different age group. For example, is an immunosuppressed child vaccinated for influenza more exposed to Covid-19 than an immunosuppressed adult, or is there no difference? The authors could add a supplementary table in relation to this data.

Author Response

Barcelona, 29th of August 2022

Dear Editor and reviewer

It is with excitement that we submit a revised version of the manuscript jcm-1871506 “Does Influenza vaccination reduce the risk of contracting COVID-19? “ 

We appreciate the comprehensive comments and feedback from the reviewers. The revisions, based on the co-authors’ collective input, have carefully considered all comments and includes several changes which we agree benefits the manuscript. We have responded specifically to each suggestion below.

Best wishes,

Francesc Alòs

Reviewer 2 Report

This is a very relevant study that contradicts some of the previously presented data claiming a clear effect. I miss in the discussion the comparison with the vaccines used by the other authors in order to better understand the differences in the results. It would be useful for the reader to know the impact of nasally administered flu vaccine considering the stimulation of the local innate immunity in addition to vaccination against flu. The discussion in this sense would increase the interest and novelty of the present document.

Author Response

(The authors gave the same response as above.)
